

# Under pressure—exploring partner changes, physiological responses and telomere dynamics in northern gannets across varying breeding conditions

David Pelletier[1,2], Pierre U. Blier[1], François Vézina[1], France Dufresne[1], Frédérique Paquin[1], Felix Christen[1] and Magella Guillemette[1]

[1] Department of Biology, Université du Québec à Rimouski, Rimouski, Québec, Canada
[2] Department of Biology, Cégep de Rimouski, Rimouski, Québec, Canada

Corresponding author
David Pelletier, david.pelletier@cegep-rimouski.qc.ca

## ABSTRACT

**Background**. Life history theory predicts trade-offs between reproduction and survival in species like the northern gannet (*Morus bassanus*). During breeding, demanding foraging conditions lead them to expand their foraging range and diversify their diet, increasing the risk of reproductive failure. Changing partners may enhance breeding success but lead to more physiological costs.

**Methods**. To investigate the physiological costs of reproduction upon partner changes, we measured and compared 21 biomarkers related to telomere dynamics, oxidative stress, inflammation, hematology, nutritional status, and muscle damage. We used a longitudinal approach with gannets ($n = 38$) over three contrasting years (2017, 2018 and 2019).

**Results**. Our results suggest that annual breeding conditions exert a greater influence on physiological changes than partnership status. Individuals that changed partner experienced greater short-term stress than retained partners. This transient increase in stress was marked by short-term increases in oxidative lipid damage, lower antioxidant capacity, signs of inflammation, and greater weight loss than individuals that retained partners. During favorable conditions, individuals that changed mates had stabilized telomere length, decreased antioxidant capacity, glucose concentration, and muscle damage, along with increased oxygen transport capacity. Conversely, unfavorable breeding conditions led to increased telomere attrition, stabilized antioxidant capacity, decreased inflammation susceptibility, diminished oxygen transport capacity, and increased muscle damage. In the cases where partners were retained, distinct physiological changes were observed depending on the year's conditions, yet the telomere dynamics remained consistent across both partnership status categories. During the favorable year, there was an increase in unsaturated fatty acids and oxygen transport capacity in the blood, coupled with a reduction in inflammation potential and protein catabolism. In contrast, during the unfavorable year in the retained mates, we observed an increase in oxidative DNA damage, antioxidant capacity, weight loss, but a decrease in inflammation susceptibility as observed in changed mates.

**Discussion**. Our study shows that behavioral flexibility such as mate switching can help seabirds cope with the challenges of food scarcity during reproduction, but these coping strategies may have a negative impact on physiological status at the individual level. In addition, the marked reduction in telomere length observed during harsh conditions,

coupled with the stabilization of telomere length in favorable conditions, highlights the long-term physiological impact of annual breeding conditions on seabirds. These findings underscore the effect on their potential survival and fitness, emphasizing that the influence of annual breeding conditions is greater than that of partnership status.

## INTRODUCTION

According to life history theory, limited resources create energy allocation trade-offs between different components of fitness (*Stearns, 1992*). In times of food scarcity, increasing reproductive investment can hinder survival. One central question in life history theory is how these trade-offs occur at the organismal level (*Stearns, 2000*). Although the negative relationship between reproduction and future reproduction or lifespan, commonly referred to as the ''cost of reproduction'' (*Williams, 1966*), is well-established, the mechanistic link between the two remains a conundrum in the field (*Flatt & Heyland, 2011*). Understanding this trade-off from a physiological perspective is essential as it holds the key to unraveling the underlying processes that govern survival and reproductive strategies.

It is possible that such trade-offs occur because increased reproductive effort cause damage to the soma (growth) or suppress repair and maintenance mechanisms (*Salmon, Marx & Harshman, 2001*; *Wang, Salmon & Harshman, 2001*). It has been suggested that oxidative stress, characterized by an imbalance between the production of reactive oxygen species (ROS) and the organism's antioxidant capacity to neutralize them, may be a primary mechanism in modulating short- and long-term life history trade-offs (*Monaghan, Metcalfe & Torres, 2009*). This imbalance leads to damage to lipids, proteins, or DNA (*Halliwell & Gutteridge, 2015*), and has been linked to effects such as tissue degradation caused by increased investment in reproductive-related activities (*e.g.*, protection, feeding, *etc.*). Such changes may influence reproductive performance, growth patterns, senescence, and survival (*Costantini, 2008*; *Costantini & Bonadonna, 2010*; *Costantini et al., 2014*; *Lin et al., 2022*; *Metcalfe & Alonso-Alvarez, 2010*; *Montoya et al., 2016*; *Williams, 2012*).

However, the relationship between parental effort and oxidative stress has been subject to debate and contrasting findings. While some studies have found evidence that high levels of parental effort induce oxidative stress (*Guindre-Parker & Rubenstein, 2018*; *Norte et al., 2010*), others have reported no relationship (*Beaulieu et al., 2011*; *Wegmann, Voegeli & Richner, 2015*). This controversy extends beyond the types of physiological traits measured (see *Fowler & Williams, 2017*; *Hegemann et al., 2013*) and is further complicated by differing interpretations and hypotheses, including the oxidative shielding (*Blount et al., 2016*) and hormesis hypotheses (*Alonso-Alvarez, Canelo & Romero-Haro, 2017*). These differing views contribute to an intricate and nuanced understanding of the cost of reproduction
(*Speakman & Garratt, 2014*), particularly when considering the effects of oxidative stress across different species.

At the intraspecific level, most of the studies has been done with small short-lived passerines, that compromise oxidative defenses when faced with increased reproduction workload (*Alonso-Alvarez et al., 2004*; *Wiersma et al., 2004*). According to the evolutionary theory, long-lived seabirds should avoid long-term impact of workload and oxidative stress on adult survival and therefore sacrifice current reproduction in prioritizing self-maintenance. They should therefore exhibit higher oxidative defenses, and suffer less from oxidative stress than short-lived species (*Xia & Møller, 2018*) in response to increased breeding effort. Indeed, long-lived birds have high levels of nonenzymatic antioxidants (*Falnes, Klungland & Alseth, 2007*; *Finkel & Holbrook, 2000*; *Galván et al., 2012*) (but see *Cohen et al., 2008*).

Long-lived seabirds also exhibit contrasting telomere dynamics when compared to small, short-lived passerines (*Haussmann et al., 2003*; *Tricola et al., 2018*). Telomeres, the protective envelopes at the chromosomes-ends (*Blackburn, 1991*), not only play a crucial role in cellular aging and lifespan (*Lopez-Otin et al., 2013*), but their attrition is considered as a biomarker of cumulative stress (*Bateson, 2016*) and an indirect reflection of individual quality (*Angelier et al., 2019*). Unlike small passerines that experience a rapid decline in telomere length over time as a consequence of the trade-off between reproduction and somatic maintenance (*Monaghan, 2014*), long-lived seabirds display decreasing, relatively stable or even increasing telomere lengths with age (*Bauch, Becker & Verhulst, 2013*; *Haussmann et al., 2003*). This intriguing pattern suggests a distinctive strategy employed by these seabirds to sustain cellular integrity and achieve longevity. For instance, *Heidinger et al. (2012)* observed an age-dependent increase in telomere length in the wandering albatross (*Diomedea exulans*), potentially attributed to elevated investment in self-maintenance and enhanced resistance to oxidative stress. The divergent telomere dynamics observed in long-lived seabirds underscore the intricate interplay of lifespan and reproductive strategies across species.

Divorce has rarely been considered in studies based on life history theory but starting a relationship with a new partner in a long-lived species with biparental care presents its own set of challenges and potential costs (*Black, 1996*), such as costs for searching for a new mate, fighting rivals, changing for a poorer mate, and costs related to an initial inefficiency in reproduction with a new mate. We recently showed that individual northern gannets (*Morus bassanus*, hereafter "gannets") that switch partners must increase their parental effort by increasing their feeding effort during unfavorable reproductive years, when individuals are under nutritional stress (*Pelletier et al., 2023*). These consequences may increase physiological stress, particularly if an individual of an obligate biparental care species must compensate for decreased parental effort by their partner (*Houston, Székely & McNamara, 2005*). However, little is known about the costs and physiological impacts of these partner changes.

Thus, to delineate the relationship between reproductive investment (as partnership change) and physiological costs of reproduction, we measured and compared 21 physiological biomarkers between gannets that stayed with previous mates and others

that changed. We used longitudinal tests (for intraindividual comparison) with data recorded in gannets during three subsequent and highly contrasted years in terms of population breeding success and diet. We evaluated partnership status with a long-term ringing monitoring program. We sampled blood during incubation and chick-rearing periods to measure biomarkers of physiological condition: telomere dynamics (relative telomere length, telomere rate of change in red blood cells), oxidative damage (thiobarbituric acid reactive substances, 8-hydroxy-2′-deoxyguanosine), antioxidant capacity, oxidative susceptibility (with fatty acid profiles and peroxidation index), inflammation (inflammation susceptibility with $\omega6/\omega3$ ratio and inflammation consequence with plasma albumin and globulin concentration), hematological indicator of stress (heterophils/lymphocytes ratio), oxygen transport capacity (hematocrit), body mass, nutritional status (plasma concentration in glucose, triglycerides, beta-hydroxybutyrate, total protein, uric acid), and muscle damage (creatine kinase activity).

Our paper aimed to assess the relationship between partnership status and stress across years with contrasted breeding conditions and improve our understanding of the intrinsic mechanisms implied in the trade-off process between reproduction and survival. We hypothesized that behavioral flexibility, expressed as mate change, represent an important cost and challenges the physiological status of adults. Secondly, we hypothesized that breeding conditions modulate the physiological cost of mate change. Therefore, we predicted that individuals that change partners will have higher rate of telomere attrition, higher levels of stress, increased oxidative damage, decreased plasma total antioxidant capacity, increased oxidative damage susceptibility, more inflammation, decreased oxygen-carrying capacity, increased weight loss, higher nutritional stress, and increased muscle damage.

## MATERIALS & METHODS

### Study site and species

We conducted fieldwork on Bonaventure Island (48°30′N, 64°09′W) located in the Parc national de l'Île-Bonaventure-et-du-Rocher-Percé in the Gulf of St. Lawrence (Québec, Canada), the largest gannet colony in Canada, where a long-term study started in 2008. This colony was monitored annually from 2008 to 2019 for partnership status and breeding success. Gannets in this population are known to exhibit a highly variable divorce rate ($22 \pm 3\%$, range: 13–46%), which is influenced by the previous year's breeding success. During the same period, breeding success remained low and variable, ranging from 3% to 60%, with an average of $32 \pm 5\%$ (*Pelletier & Guillemette, 2022*).

### Fieldwork

Between 2017 and 2019, we captured gannets once or twice during the breeding season (incubation and chick rearing periods) using a noose pole. We recorded the body mass (BM) of the birds, and when two BM measurements were taken over the season, we calculated the body mass variation (BMvar, averaging $27 \pm 7$ days between two measures). All bird capture and handling methods were approved by the Animal Care Committee (ACC) of the Université du Québec à Rimouski and complied with the guidelines of

the Canadian Council on Animal Care (CCAC; CPA-49-12-102, CPA-65-16-177). We marked each bird with a U.S. Fish and Wildlife Service steel ring with an alphanumeric code and colored plastic band (permit numbers SC25, RE-27, 10704) with approval from Société des établissements de plein air du Québec (permit numbers PNIBRP-2008-001 to PNIBRP-2019-001). We identified established partnerships in the set of monitored nests as previously described (*Pelletier et al., 2023*; *Pelletier & Guillemette, 2022*).

We plucked breast feathers (for sex determination) and we took blood samples (less than 5 ml, approx. 0.16% volume/weight) as described in *Pelletier et al. (2023)*. In the field, blood samples were divided in two tubes, one for blood smears and hematocrit preparation, and another one was centrifuged (5 min; 1,700× g). Two capillary tubes were centrifuged during 5 min at 1,300× g for hematocrit reading. Two or three blood smears were prepared in the field using the two-slide wedge technique, fixed 2 min in absolute methanol and air dried prior to storage (*Clark, Boardman & Raidal, 2009*). From the centrifuged tube, plasma and blood cells phases were divided in cryotubes and we kept them in liquid nitrogen (−196 °C) for up to 6–7 days and sorted them on dry ice (−78 °C) before storing them in a −80 °C freezer until biomarkers analyses. The birds were immediately released near their nesting site after blood sampling. Handling time was kept to a maximum of 15 min and the blood sample was taken within the first 5 min. None of the birds displayed any immediate harmful effects.

## Telomere length

Red blood cells were centrifuged from blood samples and stored previously at −80 °C. DNA was extracted with DNeasy Blood and Tissue kit (Qiagen, Hilden, Germany). Agarose gel electrophoresis was used to determine the integrity of genomic DNA samples and DNA was checked using the ratio of absorbance ($A_{260}/A_{280}$) on a NanoDrop 2000 (Thermo Fisher Scientific, Waltham, MA, USA). The telomere length assay was adapted from the published original protocol (*Cawthon, 2002*), from the modified protocol for birds (*Criscuolo et al., 2009*), and from the qPCR protocol developed for the Roche LightCycler 480 (LC480) (*Jodczyk et al., 2015*). This approach allows for the relative quantification of the mean telomere length in blood cells by assessing the quantity of telomere repeats (T) within a sample of genomic DNA, in relation to the amount of a single copy reference gene (S). This relationship is then expressed as a T/S ratio. For the telomeric DNA and the single copy reference gene, qPCR was conducted in duplicate and in isolated reactions using the LC480. Care was taken to ensure that the T and S reactions for each individual sample maintained the same relative position within the 96-well plate format. Primers for the telomere PCR were telg (5′-ACACTAAGGTTTGGGTTTGGGTTTGGGTTTGGGTTAGTGT-3′) and telc (5′-TGTTAGGTATCCCTATCCCTATCCCTATCCCTATCCCTAACA-3′) (*Cawthon, 2009*). Primers for the single copy reference gene GAPDH (glyceraldehyde-3-phosphate dehydrogenase) (*Atema, Oers & Verhulst, 2013*) were GAPDH-F$_{sula3}$ (5′-CCTGTTCATCACTGCCAGGT-3′) and GAPDH-R$_{sula3}$ (5′-AGCATCCCATACACCCTCCA-3′). The GAPDH primers were designed by Primer Premier 5.0 (Premier Biosoft International, Palo Alto, CA, USA). Amplicon lengths were 160 bp, the GC content was 55%, and the melting temperature was between 61.13 °C and 62.63 °C. We compared the

primer sequences to known gene databases at NCBI, using the BLAST option. BLAST of the 2 primers, primer F and primer R identified GAPDH gene of *Sula nebouxxi* (blue-footed booby) with *E*-value of 0.33 or less. All the primers were synthesized and purified by Sigma Life Science (Ontario, Canada). We performed qPCR for both GAPDH and telomeres using 20 ng of DNA per reaction (*Criscuolo et al., 2009*; *Jodczyk et al., 2015*). The telomere primers telc/telg were used at a concentration of 300 nM and primers GAPDH-F$_{sula3}$/GAPDH-R$_{sula3}$ at 900 nM for a final volume of 20 µl containing 10 µl of LightCycler$^{®}$ 480 SYBR Green I qPCR Master Mix.

The thermal cycling profile for the telomere PCR consisted of the following steps: 95 °C for 12 min followed by 40 cycles (3-step) of 95 °C for 10 s, 62 ° C for 10 s and 72 °C for 5 s. GAPDH PCR started with 12 min at 95 °C followed by 40 cycles (3-step) of 95 °C for 10 s, 58 °C for 10 s and 72 °C for 6 s. Melt curve analysis was performed at the end of each run to verify the specificity of PCR amplification products. Each 96-well plates included non-target controls and serial dilutions (with 80 ng, 20 ng, 5 ng, 1.25 ng and 0.3 ng of DNA per well), run in triplicate, of DNA from the same reference bird (the internal calibrator), which was used to generate a reference curve to control for the amplifying efficiency of the qPCR and to calibrate the threshold Ct. Mean values per plate were used to calculate relative T/S ratios of target individual relative to the reference individual using the formula $2^{-\triangle\triangle Ct}$ , where $[2^{C_t(telomeres)}/2^{C_t(GAPDH)}]^{-1}$. T/S ratios were used to estimate the average relative telomere length (TL) and $TL_{year1}-TL_{year0}$ for the annual telomere rate of change (TROC). Efficiencies of qPCR reactions were between 96–113% for the five telomeres plates (T plates) and between 98–107% for the five GAPDH plates (S plates). Mean intraplate coefficient of variation was $0.5 \pm 0.7\%$ for the Ct values of GAPDH assays and $0.9 \pm 0.7\%$ for the Ct values of telomere assays, and interplate coefficient of variation was 3.8% (T plates) and 2.2% (S plates) for $\triangle C_t$ values of the reference sample. Intraclass correlation coefficient for intra- and inter- runs (ICC; based on 2 samples repeated over the 96-well plates) were 0.975 and 0.929 for the five T plates, and 0.968 and 0.910 for the five S plates, respectively (*Eisenberg, 2016*; *Verhulst et al., 2016*).

## Oxidative stress and inflammation

We measured two biomarkers commonly used of oxidative damage (thiobarbituric acid reactive substances (TBARS) and 8-hydroxy-2′-deoxyguanosine (OHdG)), one biomarker of total antioxidant capacity (TAC) in plasma, two biomarkers of peroxidation susceptibility (peroxidation index of plasma (PI$_p$) and blood cells (PI$_c$)) and three biomarkers of inflammation susceptibility ($\omega6/\omega3$ ratio in plasma and blood cells, and plasma globulins (GLOB) concentration, see next subsection). The method for measuring TBARS and TAC in gannets has been described elsewhere in depth (*Pelletier et al., 2023*). We corrected TBARS for triglycerides concentration in plasma (TBARSt) (*Perez-Rodriguez et al., 2015*).

Oxidative DNA damage was assessed by measuring the amount of OHdG in plasma, using commercial kits (DNA/RNA oxidative damage ELISA Kit; Cayman Chemical, Ann Arbor, MI, USA). We used kit instructions for measurement of OHdG in plasma (in triplicate) with a dilution of 1:50 (10 µL of plasma into 490 µL of ELISA buffer). The 96-well plates were read in EnVision Multimode Microplate Reader at a wavelength of 412

nm. Mean $\pm$ SD intraplate coefficient of variation was $4.2 \pm 3.9\%$ and interplate coefficient of variation was 5.0%.

$PI_p$ and $PI_c$ were determined for plasma and blood cells . Fatty acid (FA) transmethylation protocol was adapted from *Lepage & Roy (1984)* and *Ekström et al. (2017)* for plasma and blood cells. We performed trans-methylation of 50 µL of plasma (or blood cells) with 500 µL of toluene, 3 ml of a 12% sulfuric acid ($H_2SO_4$) methanol solution directly in test vials, at 90 °C in a dry bath for 1 h. Vials were vortexed before the dry bath, after 30 min and at the end of the procedure and were subsequently cooled at room temperature. At the end, we added 3 ml of nano water and 500 µL of hexane (for the fatty acids methyl ester (FAME) extraction), followed by centrifugation ($3,000 \times g$) for 10 min at room temperature. We collected the supernatant, containing FAME in toluene-hexane, in a test tube. Using gas chromatography (Trace Ultra 100, Thermo Fisher Scientific, Waltham, MA, USA) equipped with a 60 m $\times$ 0.25 mm i.d. capillary column (DB-23, Agilent Technologies Canada, Mississauga, ON, Canada), we separated and quantified the FAMEs. We utilized helium as the carrier gas, maintaining a constant pressure of 230 kPa, and set the vaporization temperature at 230 °C with a split injection of 100 ml min$^{-1}$. The programming began at 50 °C, then rose to 140 °C at a heating rate of 25 °C min$^{-1}$, followed by an increase to 195 °C at a heating rate of 3 °C min$^{-1}$, and finally increased at a rate of 4 °C min$^{-1}$ up until 225 °C, which we maintained for 5 min. We identified the individual methyl esters by comparing them with known standards (Supelco 37 Component FAME mix). The proportion of each FA was calculated by dividing each peak area with the sum of all FA peak areas in each individual. Proportions of $\omega$6 and $\omega$3 fatty acids were cumulated to evaluate inflammation susceptibility in plasma and blood cells. PI, was calculated with cumulated percentages of unsaturated fatty acids per categories according to *Hulbert et al. (2007)* (values without units):

$$PI = \sum [(0.025 \times \% \text{ monoenoics}) + 1 \times (\% \text{ dienoics}) + 2 \times (\% \text{ trienoics})$$
$$+ 4 \times (\% \text{ tetraenoics}) + 6 \times (\% \text{ pentaenoics}) + 8 \times (\% \text{ hexaenoics})]$$

where % monoenoics = cumulated percentage of monounsaturated fatty acids (MUFA, with one double bond) and % dienoics, trienoics, tetraenoics, pentaenoics, hexaenoics are cumulated percentages of PUFA with two, three, four or six double bonds, respectively.

## Hematological biomarkers

The fixed and dried blood smears were stained during the weeks following blood sampling using a Wright-Giemsa modified staining protocol (*Woronzoff-Dashkoff, 2002*). Blood smears were examined by only one observer with a compound microscope at 400 and $1,000 \times$ magnification. Leukocytes were identified as heterophils (H), lymphocytes (L), eosinophils, monocytes, or basophils (*Campbell, 2015*). At least 100 white blood cells were counted per slide in duplicates or triplicates. We calculated HL ratios as a hematological index of stress (*Davis, Maney & Maerz, 2008*). This ratio has been shown to scale positively with plasma levels of corticosterone, the primary vertebrate stress hormone (*Davis, Maney & Maerz, 2008*; *Gross & Siegel, 1983*; *Ots & Horak, 1996*; *Vleck et al., 2000*).

Means ± SD intraindividual coefficient of variation for H count, L count and HL ratio was 6.3 ± 4.9%, 9.0 ± 7.0% and 12.3 ± 8.9%, respectively. Hematocrit (HCT) was determined in the field (in duplicates) after centrifugation. HCT was divided by plasma total protein concentration to remove the effect of hemoconcentration when there is dehydration (possibly caused by elevated temperatures at colony) and subsequent artificial increase in HCT. Mean ± SD intraindividual coefficient of variation was 0.4 ± 0.9%.

## Nutritional status and muscle damage

We evaluated nutritional status (fasting/starvation levels) by measuring glucose (GLU), triglycerides (TRIG), uric acid (URIC), total protein (TP), albumin (ALB), and globulins (GLOB) concentrations in plasma and ALB/GLOB ratio (A/G) with IDEXX Catalyst Dx™. In addition, the study of variations in ALB, GLOB and A/G ratios is also important for health evaluation in birds. For example, a decrease in the A/G ratio may indicate malnutrition, inflammatory conditions, immune response or liver dysfunction, while a decrease may indicate dehydration or immunodeficiency (*Lumeij, 2008*). We assessed muscle damage by determining creatine kinase (CK) activity in plasma using the same device. Using a commercial colorimetric assay kit (Beta-Hydroxybutyrate (Ketone Body) Colorimetric Assay Kit, Cayman Chemical Assay, USA) and an EnVision Multimode Microplate Reader (with readings at 450 nm), we also measured beta-hydroxybutyrate (BHB), a ketone body. This serves as a marker of fatty acid catabolism during fasting (*Vleck & Vleck, 2002*). *Pelletier et al. (2023)* detailed these methods.

## Data analysis
### Partnership status

We used the same dataset of partnership status of *Pelletier & Guillemette (2022)*. We determined three categories for partnership status: retained mates, divorced mates, and lost mates. However, in this study, we utilized only two categories: "retained mates" and "changed mates". In the "changed mates" category, we combined lost and divorced mates, primarily because of the low number of lost mates and our interest in initially exploring the effects of parental effort and stress following mating with a new partner.

### Statistical analysis

We conducted all statistical analyses using the statistical software R 4.0.3 (*R Core Team, 2020*) and we made plots with 'ggplot2' package (*Wickham, 2016*).

We computed longitudinal tests using the same group over time (intraindividual comparison) and the annual difference for each predictor was used as response variable. We analyzed the effect of these predictors (categorical, fixed factors): partnership status (changed *versus* retained mates), annual environmental condition for breeding (indicated by the predictor "year"), breeding stage (incubation (or empty nest) and chick rearing)), and sex on various continuous response variables (biomarkers) using multiple univariate linear mixed models. We used the function "lmer" in the package 'lme4' to run mixed-effects models (*Bates et al., 2015*) and we also log-transformed and standardized (z-scored) continuous response variables to meet statistical test assumptions and improve comparability with other studies (*Verhulst, 2020*). We checked for outliers

with the 'rosnerTest' function in the 'EnvStats' package (*Millard, 2013*). We systematically tested each biomarker for the effect of julian date and body mass, and if the relationship was significant, we used the residuals. Thus, we extracted and used residuals to remove the effect of Julian date for BM and BMvar, considering the effect of decreasing BM during breeding season (*Gebhardt-Henrich et al., 1998*; *Hillstrom, 1995*; *Merila & Wiggins, 1997*). Also, we removed the effect of BM on TRIG. Then, for each response variable, we built 64 candidate models with the function "dredge" in the package "MuMIn" (*Barton, 2009*): a null model (only with bird ID as a random factor) and models including partnership status, year, breeding stage, sex and interactions with this global model call: biomarker ~(1|ID) + Partnership status × Year + Partnership status × Breeding stage + Sex. We added URIC as covariate in models for TAC, knowing that they might be correlated (*Cohen, Klasing & Ricklefs, 2007*; *Costantini, 2011*). We calculated an Akaike information criterion value corrected for small sample (AICc) (*Burnham & Anderson, 2002*), and we chose the most parsimonious model including partnership status (first objective of the study) with the lowest AICc. Model including partnership status with unreasonable $\Delta$AICc ($>2.5$) were excluded (see Table S1). We used a likelihood ratio test (using the Chi square distribution) of the best model compared to a null model to determine if set of predictors retained in the best model were explaining variation of each dependent variable. Considering the large number of univariate statistical tests, we controlled type I errors using the false discovery rate control based on the Benjamini–Hochberg procedure applied to the $p$-values of the selected models (*Benjamini & Hochberg, 1995*). Estimated marginal means of annual differences in biomarkers were considered significant when their confidence intervals did not include zero (= average of log-transformed data) and when the $p$ value calculated by the "summ" function ("jtools" package, (*Long, 2022*)) was $<0.05$ (Table S2). With this function, models were fitted using the restricted maximum likelihood (REML) approach and $p$ values were derived using the Satterthwaite approximations. Normality of residuals and homogeneity of variances was confirmed visually for all models. Since the sub-samples were unbalanced between the categories of predictors used in the models, we conducted post hoc tests with the Tukey method for comparing groups of estimates with estimated marginal means using the function "emmeans" ("emmeans" package, (*Lenth, 2023*)). In addition, we computed the effect size in Table S2 by calculating Cohen'$d$ (*Cohen, 1988*) and using these categories: $d = 0.1$ for very small effect size, $d = 0.2$ for small effect size, $d = 0.5$ for medium effect size, $d = 0.8$ for large effect size, $d = 1.2$ for very large effect size, and $d = 2.0$ for huge effect size (*Sawilowsky, 2009*).

As described in *Pelletier et al. (2023)*, we used gannets' breeding success as proxy of annual environmental conditions where high breeding success was related to abundance and availability of their main prey (mackerel, herring, and capelin). Then, we qualified 2018 as a "good or favorable year" (breeding success: 60%), 2019 as "bad or unfavorable year" (breeding success: 13%), and 2017 as a "medium or intermediate year" (breeding success: 41%). It should be noted that the breeding success recorded in 2018 was the best recorded since monitoring began in 2008, and that the 2019 success was the second-worst recorded.

All values are reported as mean ± standard error of the mean (SEM) in the text and mean ± 95% confidence intervals (CI) in the figures.

## RESULTS

Thirty-eight northern gannets (17 females and 21 males) were studied for two (25) or three years (13) between 2017, 2018 and 2019 to evaluate the influence of partnership status and annual breeding conditions on 21 attributes of telomere length, oxidative stress, inflammation susceptibility, hematology, body mass, nutritional status, and muscle damage and (see Tables S1 and S2 for the summary of the linear mixed models used and Table S3 for the detailed results). Partnership status varied between years: 12% (2/17) changed their partner in 2017, 18% (7/38) in 2018, and 43% (15/35) in 2019.

The longitudinal analyses showed that the predictor "year" influenced 16 biomarkers (TL, TROC, OHdG, TAC, $\omega6/\omega3_c$, HCT/TP, BMvar, GLU, BHB, TP, ALB, GLOB, A/G, URIC, CK), "breeding stage" influenced six biomarkers (TROC, TBARSt, HCT/TP, BM, GLU, BHB), and "partnership status" influenced five biomarkers (TBARSt, TAC, BMvar, TP, GLOB) (see Fig. 1 and Table S2).

### Telomere length

TL did not vary with partnership status ($F_{1,76} = 0.62$, $p = 0.621$), but varied with year ($F_{2,67} = 20.75$, $p < 0.0001$). TL was similar in 2017 and 2018 ($t_{67} = -1.80$, $p = 0.180$), but TL was 137% higher in 2017 ($t_{69} = 3.29$, $p = 0.004$) and 279% higher in 2018 compared to 2019 ($t_{52} = 6.42$, $p < 0.0001$). The annual TROC varied only with year ($t_{12} = -6.69$, $p < 0.0001$) and breeding stage ($t_{34} = 2.06$, $p = 0.047$) but not with partnership status ($t_{16} = 0.24$, $p = 0.812$). TROC was lower during the unfavorable year, with a huge effect size (Cohen's $d = -3.86$) and was 126% higher (lower reduction in TL) during the incubation period compared to the chick-rearing period. Consequently, during a favorable year, gannets preserved their telomere length on average (Fig. 2A). Based on the biomarkers highlighted in the longitudinal analyses (Fig. 1), we explored relationships between each biomarker and the TROC using a nonparametric Spearman's rank correlation coefficient to detect trends. Our results revealed two weak correlations between OHdG and TROC (Fig. 2B, Spearman's $\rho = -0.422$, $p = 0.002$) and between HL ratio and TROC (Fig. 2C, Spearman's $\rho = -0.466$, $p < 0.001$). These correlations indicate that individuals that show greater telomere reduction are also those with greater oxidative DNA damage and a higher HL ratio in the second year of measurement in longitudinal monitoring.

### Oxidative stress and inflammation

The annual difference in TBARSt was 60% lower in retained mates ($p = 0.047$) than changed mates (Fig. 1) and 88% lower during the incubation period than the chick rearing period ($p = 0.015$, Fig. 3A). The difference in OHdG concentration between subsequent years was not affected by partnership status ($t_{37} = 0.72$, $p = 0.478$), but varied according to years ($t_{37} = 2.04$, $p = 0.049$) with higher differences between 2018 and 2019 in retained mates (Fig. 1, mean ±95% CI did not include zero, lower 95% CI = 0.257). The annual difference in TAC in plasma varied with partnership status ($t_{35} = 2.48$, $p = 0.020$), year ($t_{35}$

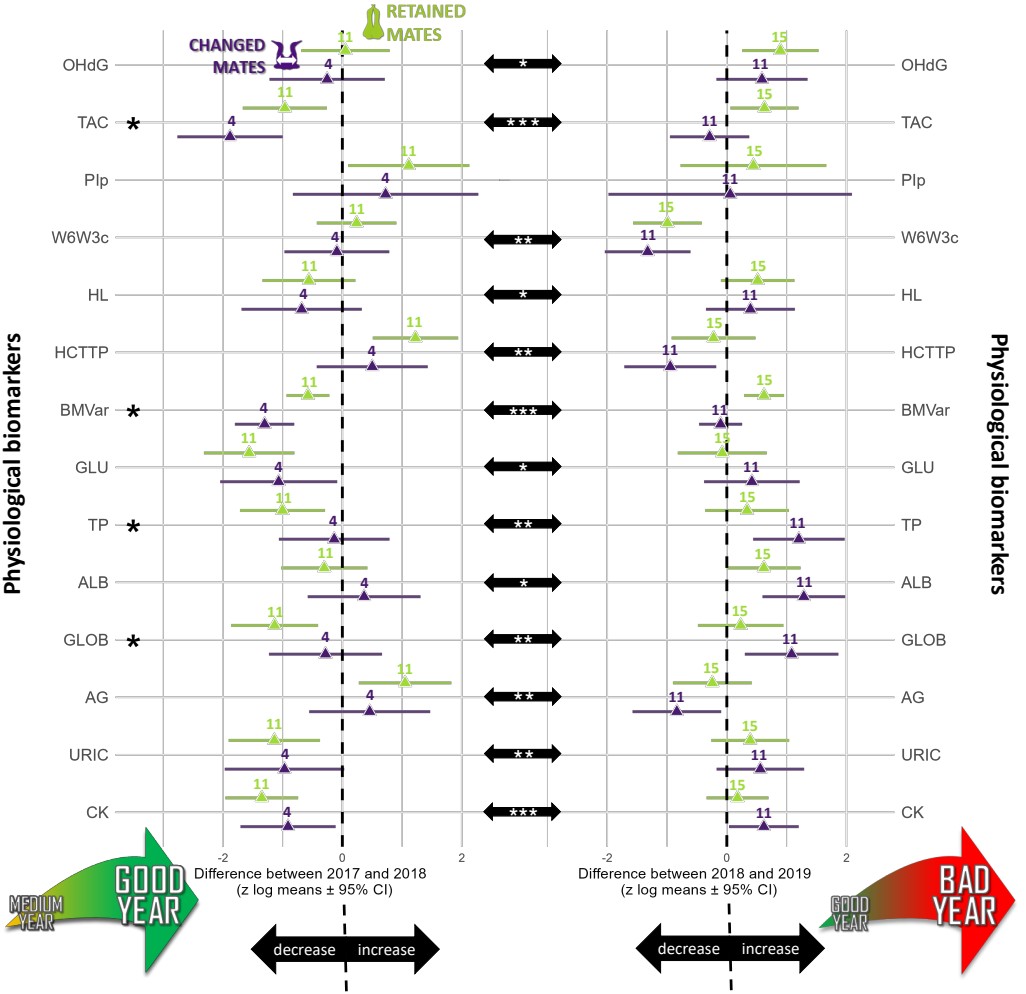

**Figure 1** **Summary of the longitudinal tests computed for annual differences calculated for each physiological biomarker studied (log-transformed and standardized) in northern gannets ($n = 38$).** This figure shows three types of comparisons: (1) the difference between partnership status (retained mates: green symbols; changed mates: purple symbols), (2) the annual differences between 2017–2018 (left column) and 2018–2019 (right column), and (3) the differences between annual differences. The asterisks on the left highlight physiological biomarkers for which there are significant differences between partnership status ($p < 0.05$). The center black arrows show significant differences between years. All possible submodels were derived from four predictors and two interactions as fixed factors and bird ID as random factor. Model selection was based on this criterion: the most parsimonious model including partnership status, with the lowest Akaike information criterion for small samples. Likelihood ratio test (using the Chi square distribution) of the best model were also used to compare the best model to a null model. Computation of $p$-values was based on t-test calculation using Satterthwaite approximation for the degrees of freedom. OHdG, plasma 8-hydroxy-2′-deoxyguanosine concentration; TAC, total antioxidant capacity of plasma; $PI_p$, peroxidation index in plasma; $W6W3_c$, omega-6/omega-3 ratio in blood cells; HL, heterophils:lymphocytes ratio; HCTTP, hematocrit divided by plasma total protein concentration; BMvar: body mass variation during the breeding season; GLU, plasma glucose concentration; TP, plasma total protein concentration; ALB, plasma albumin concentration; GLOB, plasma globulin concentration; AG, albumin/globulin ratio in plasma; URIC, plasma uric acid concentration; CK, plasma creatine kinase activity; *: $p < 0.05$, **: $p < 0.01$, ***: $p < 0.001$). Drawings credits: David Pelletier.

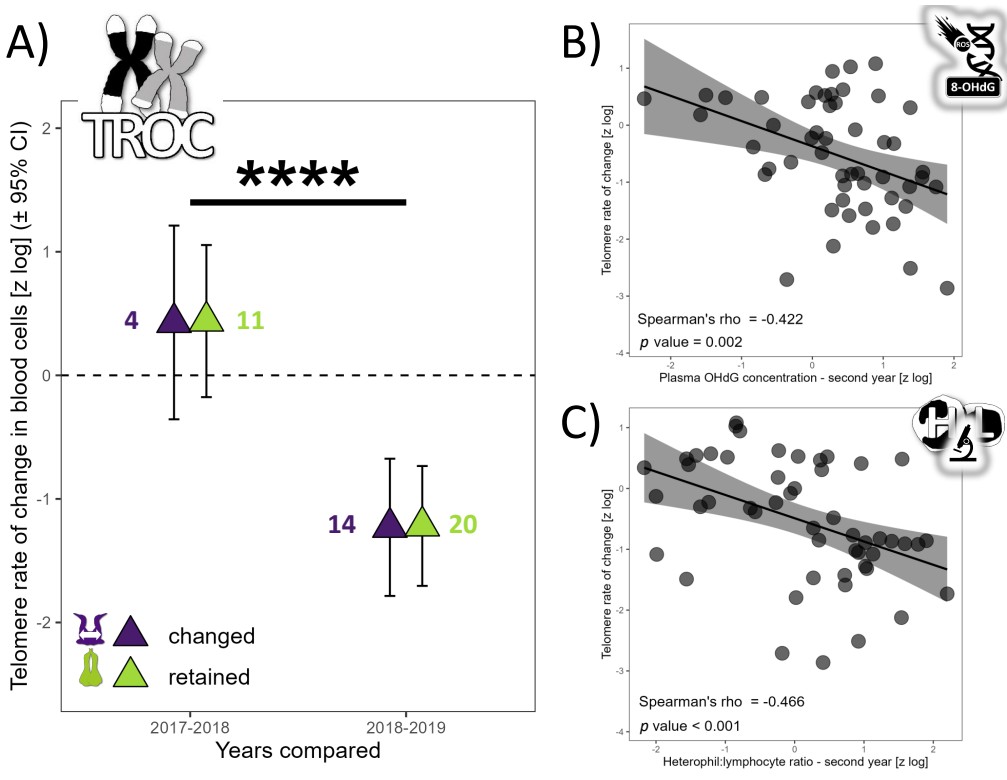

**Figure 2** **Annual telomere rate of change (TROC) in northern gannets and correlations between TROC and plasma 8-hydroxy-2′-deoxyguanosine (OHdG) concentration and heterophil:lymphocyte (HL) ratio.** (A) Annual telomere rate of change (TROC) in northern gannets ($n = 38$) that changed and retained their mates in 2017, 2018 and 2019. Results shown are estimated marginal means (±95% CI) by year obtained from a mixed linear model with bird ID as random parameter and these fixed parameters: partnership status and year. Data were log-transformed and standardized (****: $p < 0.0001$). Spearman's rank correlation coefficient (with data from longitudinal tests) between TROC and (B) plasma 8-hydroxy-2′-deoxyguanosine (OHdG) concentration measured the second year and (C) heterophil:lymphocyte (HL) ratio measured the second year (drawing credits: David Pelletier).

$= 3.92$, $p = 0.0004$) and URIC difference ($t_{35} = 2.87$, $p = 0.007$). Retained mates exhibited 151% higher TAC than changed mates, and TAC increased more in bad year (2018 to 2019, 390% more, with a very large effect size, Cohen's $d = 1.32$). TAC difference was negative in 2018 compared to 2017 for all individuals (Fig. 1), but only retained mates increased TAC in 2019 (zero not included in 95% CI, lower 95% CI = 0.060).

The annual difference in $\omega6/\omega3$ fatty acids ratio in plasma ($\omega6/\omega3_p$) was best explained by the null model (no model including partnership status was selected, because none was different from the null model ($p > 0.05$) and no model including partnership status had reasonable AICc value around 2 or less. Moreover, the annual difference in $PI_p$ did not vary according to partnership status ($t_{11} = 0.64$, $p = 0.530$) and year ($t_{11} = -1.19$, $p = 0.259$). With the blood cells, the annual difference in $\omega6/\omega3_c$ did not vary with partnership status ($t_{36} = 0.84$, $p = 0.405$) but varied with year ($t_{36} = -3.24$, $p = 0.003$, Fig. 1). The $\omega6/\omega3_c$ difference was 70% lower in 2018–2019 than 2017–2018, with a large effect size (Cohen's d

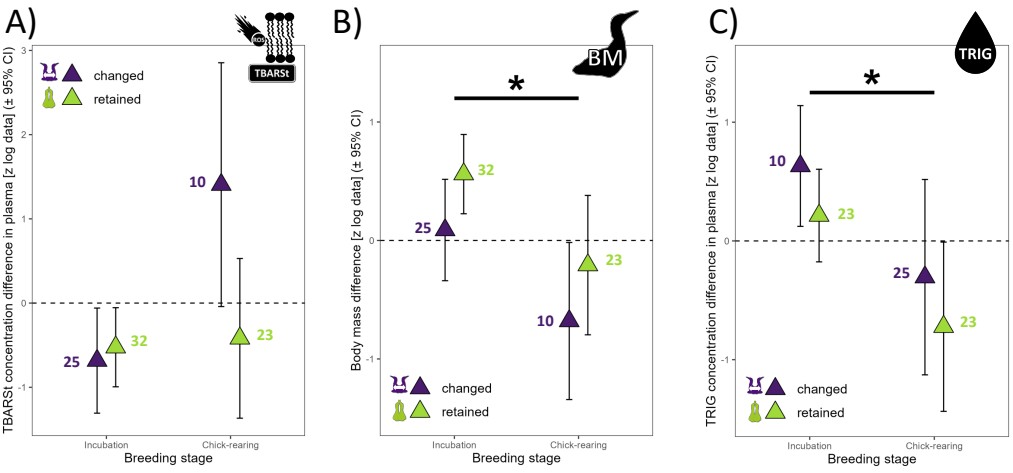

**Figure 3** **Annual differences in biomarkers measured in northern gannets ($n = 38$) for which the predictor "year" was not present in any of the best linear mixed models selected (see Table S1).** (A) Annual difference in plasma thiobarbituric acid reactive substances divided by plasma triglycerides concentration (TBARSt), (B) annual difference in body mass (BM), and (C) annual difference in plasma triglycerides concentration (TRIG). Results shown are estimated marginal means ($\pm 95\%$ CI) by breeding stage obtained from a mixed linear model with bird ID as random parameter and these fixed parameters: breeding stage and partnership status. Data were log-transformed and standardized (*: $p < 0.05$) (drawing credits: David Pelletier).

$= -1.08$). No model including the dependent variables was selected for $PI_c$, because none was different from the null model ($p > 0.05$) and no model including partnership status and year had reasonable AICc value (around 2 or less).

### Body mass

There was a trend for 60% higher annual difference in BM for retained mates ($t_{47} = 1.89$, $p = 0.064$) but all means $\pm 95\%$ CI included zero, suggesting that annual individual difference in BM was not influenced by environmental conditions (Fig. 1), but BM was 115% higher during incubation than during chick-rearing (Fig. 3B). Annual difference in BMvar was 106% higher in retained mates ($t_9 = 3.92$, $p = 0.004$) and BMvar was 231% higher during bad years (2019) compared to a good year (2018) ($t_7 = 6.87$, $p < 0.001$, with a huge effect size, Cohen's $d = 5.20$).

### Hematology

The annual difference in HL ratio did not vary with partnership status ($t_{46} = 0.28$, $p = 0.780$), but varied with year ($t_{46} = 5.46$, $p = 0.018$). There was a trend for a decrease in HL ratio between 2017 and 2018 and a trend for an increase between 2018 and 2019, but 95% CI included zero in both years (Fig. 1). The annual differences in HCT/TP where 77% lower in bad year (2019) compared to increase in a good year (2018) ($t_{42} = -3.45$, $p = 0.001$). Differences in HCT/TP tended to be lower in changed mates ($t_{42} = 1.82$, $p = 0.077$). HCT/TP increased during the favorable year for retained mates and decreased for changed mates during the most unfavorable year (Fig. 1).

## Nutritional status and muscle damage

The annual difference in GLU concentration in plasma did not vary with partnership status ($t_{43} = -1.19$, $p = 0.242$) but varied with breeding stage ($t_{43} = 2.04$, $p = 0.047$) and year ($t_{43} = 3.34$, $p = 0.002$). GLU difference increase more (+180%) during incubation compared to chick rearing period and was 341% higher in 2018-2019 than 2017-2018 (Fig. 1). During favorable years, both partnership status decreased GLU concentration in plasma (zero not included in 95% CI).

The annual difference in TRIG concentration in plasma did not vary with partnership status ($t_{43} = -1.18$, $p = 0.246$) but varied with breeding stage ($t_{43} = 2.25$, $p = 0.029$). TRIG difference was 154% higher during incubation than during chick-rearing (Fig. 3C).

No model including the partnership status was selected for the annual difference in BHB concentration in plasma, since none including this parameter had reasonable AICc (around 2 and less). The best model included breeding stage and year ($\chi^2 = 4.03$, $df = 1$, $p = 0.045$). BHB difference varied with year ($t_{15} = -2.16$, $p = 0.047$) and breeding stage ($t_{32} = 2.13$, $p = 0.041$). BHB difference was 52% lower during unfavorable year and 165% during the incubation period.

The annual difference in TP concentration in plasma varied with partnership status ($t_{42} = -2.18$, $p = 0.035$), with year ($t_{42} = 3.21$, $p = 0.002$) but did not vary with breeding stage ($t_{42} = 1.86$, $p = 0.070$). TP decreased in retained mates only during favorable years (Fig. 1) and increased during the unfavorable year in changed mates (with large effect size: Cohen's $d = 0.99$). The annual difference in ALB concentration in plasma did not vary with partnership status ($t_{43} = -1.63$, $p = 0.111$) but varied with year ($t_{43} = 2.22$, $p = 0.032$) with higher difference between 2018 and 2019 where both, retained and changed mates, increased ALB during bad year (Fig. 1). The annual difference in plasma GLOB concentration varied with partnership status ($t_{42} = -2.10$, $p = 0.042$), and year ($t_{42} = 3.20$, $p = 0.003$), and there was a trend with breeding stage ($t_{42} = 1.91$, $p = 0.063$). In the favorable year, GLOB decreased for retained mates and in the unfavorable year, GLOB increased for changed mates. The annual difference in A/G ratio did not vary with partnership status ($t_{43} = 1.35$, $p = 0.184$) but varied with year ($t_{43} = -2.90$, $p = 0.006$). In the favorable year, A/G ratio increased for retained mates and in the unfavorable year, A/G ratio decreased for changed mates.

The annual difference in URIC concentration in plasma did not vary with partnership status ($t_{43} = -0.39$, $p = 0.698$) but it varied with year ($t_{43} = 3.48$, $p = 0.001$). URIC decreased during the favorable year and increased during the unfavorable year (with a large effect size, Cohen's $d = 1.06$, Fig. 1).

The annual difference in CK activity in plasma did not vary with partnership status ($t_{43} = -1.27$, $p = 0.210$) but it varied with year ($t_{43} = 4.36$, $p = 0.0001$). In 2018, both partnership status decreased CK, but in 2019, only changed mates showed an increase in CK.

## DISCUSSION

Our study reveals a complex and nuanced interplay between pairing behavior, physiological responses, and annual breeding conditions in northern gannets, as depicted in Fig. 4. When

## A) Concerning the **PARTNERSHIP STATUS**

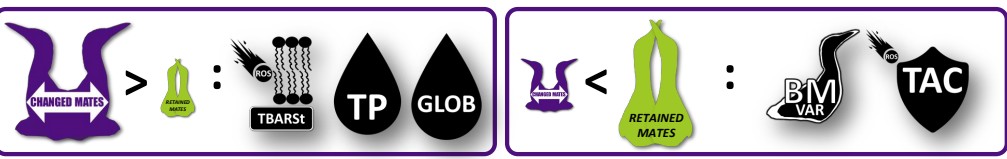

## B) Concerning the **ANNUAL BREEDING CONDITIONS**
### (for CHANGED MATES & RETAINED MATES)

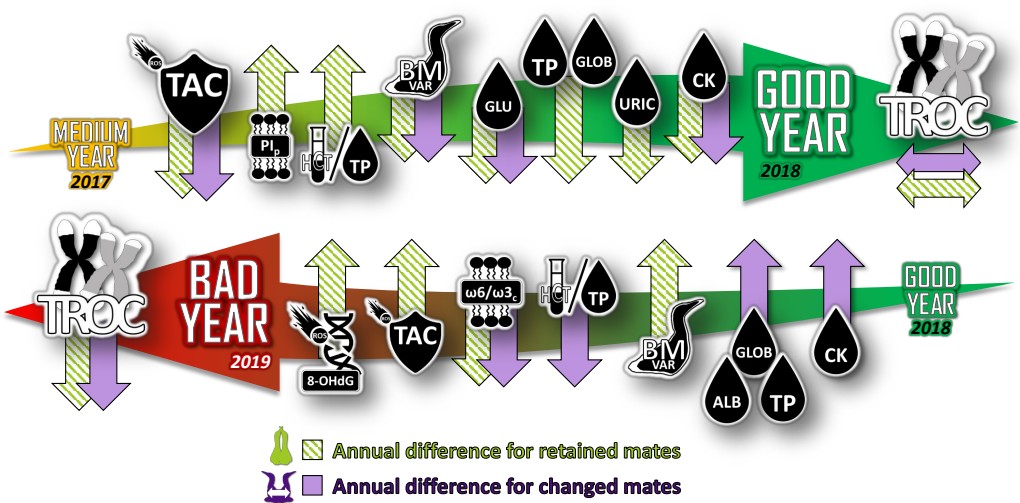

**Figure 4** Summary of the longitudinal results of physiological characteristics of northern gannets that changed their mates in 2017 (intermediate year), 2018 (good year) and 2019 (bad year). Results were obtained from mixed linear models built with bird ID as random parameter and with these fixed parameters: partnership status, year, breeding stage, sex and interactions. See Fig. 1 and Tables S1 and S2 for more details. TBARSt: plasma thiobarbituric acid reactive substances divided by plasma triglycerides concentration; TP, plasma total protein concentration; GLOB, plasma globulin concentration; BMvar: body mass variation during the breeding season; TAC, total antioxidant capacity of plasma; PIp, peroxidation index in plasma; HCT/TP, hematocrit divided by plasma total protein concentration; GLU, plasma glucose concentration; URIC, plasma uric acid concentration; CK, plasma creatine kinase activity; TROC, annual telomere rate of change; OHdG, plasma 8-hydroxy-2′-deoxyguanosine concentration; $\omega 6/\omega 3_c$, omega-6/omega-3 ratio in blood cells; ALB, plasma albumin concentration (drawing credits: David Pelletier).

transitioning from an average to a favorable year, gannets exhibit an improved physiological health regardless of their partnership status. This is marked by a stabilization of telomere length and a reduction in antioxidant capacity, weight loss, glucose concentration, and muscle damage. For individuals that remain with their previous partner, the health benefits appear even more pronounced, with enhancements in oxygen transport capacity, and reduction in both signs of inflammation and protein catabolism. However, a shift from favorable to unfavorable conditions yields the opposite effect, resulting in increased telomere attrition and decreased susceptibility to inflammation. In these unfavorable breeding conditions, the physiological response varies based on pairing behavior. Oxygen

transport capacity declines, while signs of inflammation and muscle damage escalate. Concurrently, there is an uptick in oxidative DNA damage, antioxidant capacity, and weight loss.

## Behavioral flexibility through partnership status

Divorce, as a behavioral strategy, has been demonstrated to mitigate the effects of low food abundance, subsequently enhancing breeding success in the year following a failure (*Pelletier & Guillemette, 2022*). A recent study further illuminated this phenomenon among gannets, revealing that those that paired with new partners augmented their parental effort during breeding seasons. Specifically, they increased their foraging effort, both in distance and range, during unfavorable years (*Pelletier et al., 2023*). This behavioral flexibility serves as a mechanism to counter some of the stressors precipitated by environmental fluctuations. However, these coping strategies are not without cost. They may adversely impact an individual's physiological condition and, consequently, affect future reproductive success. In our study, we provide a detailed quantification of these physiological costs, shedding a light on the intricate balance between immediate benefits and potential long-term consequences.

## Telomeres and reproduction costs

Telomere length, serving as an integrative marker of an individual's life stress and stress-coping mechanisms, and individual quality (*Angelier et al., 2019*), was examined in our longitudinal study. We found no differences in telomere length between retained and changed mates, suggesting both types of parents might not differ in 'quality'. Interestingly, gannets' telomeres were stabilized during the favorable year and shortened during unfavorable breeding conditions when parental effort increased, aligning with life history theory (*Stearns, 1992*) and observations of telomere attrition being influenced by seasonal variations and non-permanent states (*Criscuolo et al., 2020*; *Tissier et al., 2022*; *Viblanc et al., 2022*). These results are also in line with other recent studies (*Pepke et al., 2022*; *Pepke et al., 2023*), which have shown that the variation due to year effects were larger than that due to all other environmental, parental or genetic factors included, suggesting that individual heterogeneity in telomere dynamics is mainly driven by annual environmental stochasticity. This may implied that increased effort during poor feeding years (*Pelletier et al., 2023*) is associated with shortened telomeres, and a reduction in life expectancy, as in black-legged kittiwakes (*Rissa tridactyla*) (*Benowitz-Fredericks et al., 2022*).

Despite observing higher levels of lipid oxidative damage and lower antioxidant capacity in birds that changed mates, no differences were found in DNA oxidative damage and telomere length between changed and retained mates. This suggests that stress due to parental compensation after a mate change might be counterbalanced by an endogenous or exogenous protective mechanism, perhaps similar to the telomerase activity described by *Haussmann et al. (2007)*, which could prevent the negative impact on telomeres. Partnership status, therefore, does not seem to affect telomere length and dynamics, a surprising result that prompts further exploration of protective mechanisms during breeding or non-breeding seasons.

Our study underscores that annual breeding conditions have more pronounced effects on telomere dynamics than re-mating behavior in gannets. We observed that a favorable breeding year stabilizes telomere length, while an unfavorable year contributes to telomere shortening. This is in line with previous research showing that environmental rearing conditions can influence both offspring and parents' telomere length (*Viblanc et al., 2020*), and a recent meta-analysis linking exposure to stressors with shorter telomeres (*Chatelain, Drobniak & Szulkin, 2020*). The results gathered across three very contrasting years substantiate these findings, emphasizing the need for future studies to further explore the relationships between telomere length, breeding conditions, and potentially unexplored biological mechanisms.

## Physiological status

Our study did not report differences in terms of stress (evaluated with the HL ratio) according to partnership status. However, within our three-years sample scheme, the HL ratio increased and lymphocytes decreased during the unfavorable year (Fig. 1), indicating higher stress during unfavorable breeding conditions. Furthermore, oxygen-carrying capacity was not related to partnership status, instead showing a relationship in the between-year comparison. In medium-vs.-good years comparison, retained mates increased HCT and, at the opposite, in good-vs.-bad years comparisons, changed mates decreased HCT. As HCT is a key determinant of oxygen transport capacity and exercise performance (*Böning, Maassen & Pries, 2011*; *Yap et al., 2018*), increased HCT during favorable breeding conditions suggests that birds are more adept at performing endurance flights than during unfavorable conditions. Birds typically increase erythropoiesis and hematocrit rates before and during migratory flights (*Fudickar et al., 2016*), or in response to an increased workload due to brood enlargement (*Hõrak, Ots & Murumägi, 1998*). The study's findings suggest that gannets may be better equipped to support metabolic activities such as foraging flights during favorable breeding conditions due to the observed increase in hematocrit levels.

Workload induced by foraging flights or protective behavior could partly drive oxidative balance. In our study, sex was not included in any of the best models produced with the set of biomarkers used, but we found that short-term plasma oxidative damage was higher (with higher TBARSt) in changed mates. However, with a medium-term biomarker test (OHdG), there was no effect of partnership status. The increase in parental effort promoted by partnership status changes (*Pelletier et al., 2023*) does not appear to destabilize the oxidative balance long enough to cause a divergence in cumulative oxidative DNA damage. The differences in DNA damage observed in longitudinal tests between intermediate-vs.-good and good-vs.-bad years comparisons suggest that environmental conditions may have a greater impact on oxidative balance from a medium- to long-term perspective. Our study suggests that increased foraging effort during unfavorable breeding conditions and after a mate change may affect the non-enzymatic antioxidant capacity of gannet plasma. This phenomenon has been observed in small passerines, such as zebra finches (*Taeniopygia guttata*), where increased parental effort led to decreased plasma antioxidant capacity and increased oxidative stress (*Alonso-Alvarez et al., 2004*).

Our study also suggests that changes in breeding conditions may lead to annual variations in antioxidant capacity, which are partially influenced by partnership status and associated parental effort. Gannets, regardless their pairing behavior, exhibited a decrease in total antioxidant capacity (TAC) in favorable years and an increase in TAC in unfavorable years (only in retained mates), indicating a flexible physiological response to cope with oxidative stress during periods of high metabolic activity and reduced food availability. Conversely, the changed mates did not adjust their non-enzymatic antioxidant capacity in the blood to cope with the increased metabolic demands during the unfavorable breeding conditions. King penguins (*Aptenodytes patagonicus*) have been shown to resist stress by upregulating endogenous antioxidant defenses (glutathione antioxidant system) and decreasing mitochondrial efficiency (*Stier et al., 2019*) and we suspect that differences in stress markers associated with partnership status in gannets may be related to regulation of antioxidant capacity.

The difference in peroxidation index in plasma ($PI_p$) decreased between medium-*vs.*-good years to good-*vs.*-bad years, suggests an increase in oxidative damage susceptibility during unfavorable years. $PI_p$ is influenced by the composition of fatty acids (FAs) obtained mainly *via* dietary sources in seabirds (*Käkelä et al., 2007*; *Käkelä et al., 2009*). In our study, changed and retained mates share the same diet, but further analysis of plasma and prey fatty acid profiles could potentially confirm this (*Budge, Iverson & Koopman, 2006*; *Iverson et al., 2004*; *Karnovsky, Hobson & Iverson, 2012*).

The ratio of omega-6 to omega-3 fatty acids, which is a marker of inflammation susceptibility, varied between years but was not affected by partnership status. In blood cells, the ratio decreased during the unfavorable year, which suggests a higher intake of fish rich in omega-3 PUFAs or fewer prey items rich in omega-6 PUFAs. Eicosanoids, which are hormone-like lipids that modulate inflammatory and immune responses, are derived from omega-6 and omega-3 fatty acids. Omega-6 fatty acids have pro-inflammatory effects, while omega-3 fatty acids inhibit inflammation (*Larsson et al., 2004*; *Rose & Connolly, 1999*). The decrease in the omega-6/omega-3 ratio during the unfavorable year (with higher intake of fish rich in $\omega$3 PUFAs like mackerel) could potentially provide some protection against the excess inflammation associated with increased metabolism.

In our investigation, we were unable to uncover definitive evidence pointing to elevated nutritional stress correlated with partnership status, as examined through the lenses of carbohydrate, lipid, and protein metabolism. Nevertheless, nuances in protein metabolism did manifest as discernible disparities in both nutritional stress and inflammation between changed and retained mates. To elucidate, total protein (TP) and globulin (GLOB) concentrations were found to be elevated in changed mates, and the levels of TP, GLOB, and albumin (ALB) were observed to rise during the unfavorable year. Intriguingly, there was an absence of variation in protein catabolism, as signified by uric acid levels, associated with partnership status. However, we did identify an escalation during the unfavorable year (*Pelletier et al., 2023*). Elevated TP and ALB levels can act as indicators of protein reserves, such as those seen during during hyperphagia before migration (*Allison, 1955*; *Jenni-Eiermann & Jenni, 1998*; *Mori & George, 1978*), but these levels can alternatively be reflective of dehydration (*Boyd, 1981*) or chronic inflammatory states, characterized by an

increase in $\alpha$, $\beta$ or $\gamma$ globulin (GLOB) concentrations and oftentimes, a decrease in ALB (*Całkosiński et al., 2016*). Our findings propose that changed mates might be subject to an escalation in inflammation, particularly during the unfavorable year, in spite of a reduction in susceptibility to inflammation (as illustrated by the ω6/ω3 results previously detailed). Experiments conducted on laboratory rats have documented declines in ALB concentration during acute inflammation and augmentations in various globulins as a response to either chronic inflammation or infection (*Całkosiński et al., 2016*). Therefore, it may be posited that the amplified GLOB levels in changed mates could have an underlying connection to chronic inflammation (*Gustafsson et al., 1994*; *Martinez et al., 2003*).

Our study also revealed that during unfavorable years, breeding gannets showed an increase in protein catabolism, as indicated by the rise in uric acid (URIC) levels (*Mori & George, 1978*). However, their levels were lower than those reported for other piscivorous marine birds (1,190–1,784 $\mu$mol L$^{-1}$, *Newman, Piatt & White (1997)*) but higher than values reported for other avian species (178–595 $\mu$mol L$^{-1}$, *Campbell & Grant (2010)*). In contrast, there was no significant variation in triglycerides (TRIG) and $\beta$-hydroxybutyrate (BHB) levels, suggesting that we cannot assess the nutritional status of gannets based on these markers alone (*Jenni-Eiermann & Jenni, 1998*). Nevertheless, plasma activity of creatine kinase (CK) indicated that annual breeding conditions influenced muscle damage in both partnership statuses, with gannets decreasing CK levels during favorable years and increasing them during unfavorable years. These results are in accordance with the highest foraging effort associated with flight (*Guglielmo, Piersma & Williams, 2001*; *Swanson & Thomas, 2007*). This observation could likely explain the increased protein catabolism observed during unfavorable years for reproduction.

The comparison of annual differences in glucose concentration (GLU) may provide insights into potential connections between changes in parental effort and impacts on telomere length, though our study did not observe a direct correlation between GLU and TL (or TROC). GLU levels are typically regulated by hormonal control and kept within narrow limits, as their maintenance is critical for the bird's central nervous system (*Braun & Sweazea, 2008*; *Totzke et al., 1999*). Our results suggest that GLU variation in plasma may be indicative of more global consequences of stress levels than nutritional status level. This hypothesis is further supported by the HL ratio results. During favorable years, GLU levels and HL ratios were lower, and they increased during unfavorable years. The toxicity of glucose is related to damage from reactive oxygen species (ROS) through the oxidation of glucose, non-enzymatic glycation of proteins, and the production of advanced glycation end (AGEs) products, which are implicated in aging (*Holmes & Austad, 1995*). Rates of ROS production and protein glycation are known to be proportional to glucose concentration (*Bunn et al., 1976*) , and modest plasma glucose levels could be beneficial in reducing its apparent toxicity. Lower GLU levels might assist birds, which typically have 2-4x higher GLU concentration than similar-sized mammals (*Holmes & Austad, 1995*), to reduce glucose toxicity. While the reduced GLU levels observed in 2018 suggest that glucose regulation could be a contributing factor to telomere length stability measured in that year, the absence of direct associations between TL and GLU in our study limits the certainty of this connection. Moreover, the increase in GLU levels observed in 2019, combined

with increases in TP and GLOB levels as signs of inflammation, could potentially explain telomere dynamics through hyperglycemia-related oxidative stress and inflammatory processes that accelerate age-related telomere attrition (*Koliada, Krasnenkov & Vaiserman, 2015*), though further research require to substantiate this relationship.

## CONCLUSION

In summary, our study sheds light on the impacts of behavioral flexibility resulting from mate change on stress regulation, nutrition, and telomere dynamics in northern gannets. We found that interannual variation in breeding conditions (related to food availability and parental effort) has the most significant impact on the birds' health and telomere dynamics at a medium-term (three years) scale. However, at the breeding season temporal scale, our results support our first hypothesis that changing mates has a negative impact on the health condition of gannets, as indicated by lower antioxidant capacity, higher rates of lipid oxidative damage associated with inflammation, and higher mass loss during chick-rearing. Additionally, we found that the negative impact of mate change occurs during an unfavorable year when environmental conditions lead to low food abundance, confirming our second hypothesis. Therefore, from favorable to unfavorable years, the physiological status of gannets decreased, and telomere erosion accelerated. Overall, our study suggests that unfavorable environmental conditions have a more significant impact on the telomere attrition of gannets than the behavioral flexibility resulting from re-pairing behavior. Nevertheless, the physiological changes related to the partnership status decision appear to last only during the breeding season following the change and do not seem to alter individual parental quality. Furthermore, our findings indicate that pairing with a new mate is unlikely to affect the survival and fitness of the parents as opposed to harsh environmental conditions during the breeding period. As a future direction, we recommend an in-depth analysis of physiological markers that could identify traits that give individuals an advantage within a population in the face of global changes. This could potentially contribute to a better understanding of the mechanisms underlying the adaptability of seabirds to environmental variability and provide insights into how to mitigate the impact of anthropogenic activities on seabird populations.

## ACKNOWLEDGEMENTS

Thanks for the extensive commitment of sampling, laboratories and analytical personnel involved in the Laboratoire d'ornithologie marine de Rimouski (LOMR) at Université du Québec à Rimouski (Sandrine Gingras, Jolanie Roy, Safouane Khamassi, Jeanne Bouchard, Selma Elfassi-Fihri, Marie-Anne Robitaille, Richard Gravel, Roxanne Turgeon, Camille Novales, Laurence Gagnon, Marie-Eve Labonté Dupras, Catherine Destrempes, Fanny May Couture Charron, Andra Florea) and at Cégep de Rimouski (Catherine Bouchard, Laury-Ann Dumoulin, Angéline Robichaud, Catherine Gloutnez, Andréa Lévesque, Isabelle Demalsy, Alexia Tremblay, Vincent Benjamin, Josiane Lavoie-Bélanger, Laura Levesque Arsenault, Zoé Fournier, Juliette Deschenaux, Carole-Anne Michaud, Rosalie Pedneault, Gabrielle Pellerin). We sincerely thank Liette Regimbald and Veronique Desrosiers for

helping us with blood sampling and analyses; Emily Cornelius for helping us with avian hematology; François Criscuolo, Cécile Ribout (from CNRS—Université de Strasbourg), Thomas Zgirski and Jean-Michel Martin for helping us with protocol development of telomere length measurement. Thanks to Parc national de l'Île-Bonaventure-et-du-Rocher-Percé and its staff for transportation and supportive collaboration. Final thanks to Denis Réale, Frédéric Angelier and Céline Audet for being part of the jury evaluating DP's PhD thesis (in which this article appears), and thanks to many anonymous reviewers for their helpful comments. The title is inspired by ''Under pressure'', a song written by Queen and David Bowie about the stress of life and the pressures of relationships.

### Funding

This work was supported by the Canadian Natural Sciences and Engineering Research Council (NSERC) discovery and equipment grants to Magella Guillemette, by the Fonds de recherche du Québec-Nature et technologies (FRQNT) Research program for college researchers (grant number: 193215) to David Pelletier, and by the NSERC Alexander Graham Bell Canada Graduate Scholarship to David Pelletier. The laboratory analysis of telomeres was funded by the BORÉAS-UQAR research group. The funders had no role in study design, data collection and analysis, decision to publish, or preparation of the manuscript.

### Grant Disclosures

The following grant information was disclosed by the authors:
The Canadian Natural Sciences and Engineering Research Council (NSERC) discovery and equipment grants.
Fonds de recherche du Québec-Nature et technologies (FRQNT) Research program for college researchers: 193215.
NSERC Alexander Graham Bell Canada Graduate Scholarship.
The BORÉAS-UQAR research group.

### Competing Interests

The authors declare there are no competing interests.

### Author Contributions

- David Pelletier conceived and designed the experiments, performed the experiments, analyzed the data, prepared figures and/or tables, authored or reviewed drafts of the article, methodology, data curation, investigation, project administration, and approved the final draft.
- Pierre U. Blier conceived and designed the experiments, authored or reviewed drafts of the article, and approved the final draft.
- François Vézina conceived and designed the experiments, authored or reviewed drafts of the article, and approved the final draft.

- France Dufresne conceived and designed the experiments, authored or reviewed drafts of the article, and approved the final draft.
- Frédérique Paquin performed the experiments, authored or reviewed drafts of the article, methodology, and approved the final draft.
- Felix Christen performed the experiments, authored or reviewed drafts of the article, methodology, and approved the final draft.
- Magella Guillemette conceived and designed the experiments, authored or reviewed drafts of the article, supervision, and approved the final draft.

## Animal Ethics

The following information was supplied relating to ethical approvals (i.e., approving body and any reference numbers):

All bird capture and handling methods were approved by the Animal Care Committee (ACC) of the Université du Québec à Rimouski (CPA-49-12-102, CPA-65-16-177), and complied with the guidelines of the Canadian Council on Animal Care (CCAC).

## Field Study Permissions

The following information was supplied relating to field study approvals (i.e., approving body and any reference numbers):

Field experiments were approved by the Canadian Wildlife Service (permit numbers SC25, RE-27, 10704) and Société des établissements de plein air du Québec)permit numbers PNIBRP-2008-001 to PNIBRP-2019-001).

## Data Availability

The raw data are available in the Supplementary File.

## Supplemental Information

Supplemental information for this article can be found online at http://dx.doi.org/10.7717/peerj.16457#supplemental-information.

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
