# Peer review of "Under pressure—exploring partner changes, physiological responses and telomere dynamics in northern gannets across varying breeding conditions"

_PeerJ, doi:10.7717/peerj.16457_

## Round 0.1 · original submission · Major Revisions

We have received two very well detailed reviews for your manuscript. Both reviewers found merits in your study and thought it will represent an interesting contribution to the field.

However, they also raised a number of important issues, most of them related to methods and data analyses, that deserve further revisions.

In particular, Reviewer 1 raised several important concerns related to the lack of clarity (and a few problems) in the description of the statistical analyses performed.

Reviewer 2 identified numerous parts of the lab methods and analyses that need to be further described/clarified. The possibility of a ‘year effect’ should also be taken into account.

Various other relevant comments regarding the abstract, introduction and discussion were provided by reviewers and should be integrated in the next version, as they will help improve the structure and clarity of your manuscript.

Reviewer 1 ·

Basic reporting

This study is a comprehensive assessment of physiological parameters measured in up to 38 northern gannets (see comment about sample size below) in relation to their partnership status across three years. The authors showed how some physiological variables changed depending on whether individuals retained mates or divorced when changing from an average to a favorable year or from a favorable year to an unfavorable year. Generally, it appears that annual environmental breeding conditions had greater effects on physiological condition than partnership status. The study is set in a relevant life-history framework of trade-offs between reproduction and survival. I am not familiar with all of the laboratory methods applied, but they appear to be informatively described or further detailed elsewhere. The manuscript is generally well-written, and it may be an interesting contribution to the field. However, I have several comments and questions that needs to be addressed.

Experimental design

No comments.

Validity of the findings

1) The description of the data analysis is somewhat unclear. For example, which variables were modelled as fixed and random effects (I can guess the latter is bird ID but it needs to be described in the Methods). Which variables were modelled as continuous or categorical? Were all variables “log-transformed and standardized” (L278), and why were they log-transformed (provide statistical and/or biological reason), and how were they “standardized”? Furthermore, which statistical distribution was used in the mixed-effect models (and was it the same for all variables)?

2) It is not described which candidate models were constructed (and how many) and what the “null model” (L280) was (an intercept model presumably?). However, you write that “we chose the most parsimonious model including partnership status (first objective of the study) with the lowest AICc” (L294). First, does this mean that partnership status was included in all models? Second, what did you do in cases where two or more models were within AICc < 2, in which case you cannot conclude which model is the best? I strongly suggest including complete AICc tables in the appendix and reporting the ∆AICc of the best model compared to the second-best model. You may need to discuss alternative models.

3) The T/S ratios will normally be distributed around some ratio close to 1-2 (depending on the reference sample). Thus, I wonder if samples with a T/S ratio > 9 are extreme outliers. It is not mentioned in the Methods if any samples or measurements were excluded due to e.g. too large differences in Ct values among triplicates, or too high Ct threshold (failed reaction), or poor DNA quality. Furthermore, one individual had missing TL measurements (data supplement), which means that sample size is not 38, but 37. This also seems to apply to several other variables with missing values. Sample sizes should be revised and reported correctly for each variable, ideally also in the relevant figures and tables (e.g. as columns in Fig. 1 and over bars in Fig. 2).

Additional comments

4) You write that “Due to the potential relationship between breeding success and telomere dynamics, we included annual breeding success in models build for telomere length (TL) and telomere rate of change (TROC)” (L282). First, you should provide a reference for the claim and explain how breeding success was quantified and modelled. But would you not expect other biomarkers to be involved in such a relationship, e.g. measures of oxidative stress that are thought to influence TROC – i.e. why was breeding success only included in models explaining variation in telomere dynamics? Furthermore, it would have been interesting to include breeding success as a measure of reproductive success and relate this to the physiological parameters, which could provide more support for your discussion of life-history trade-offs between reproduction and individual condition/survival.

5) Fig. 2b-c presents post hoc tests of correlations between TROC and any of the other variables. I suggest specifying in the Introduction why you perform these tests (and why only for TROC and not across all biomarkers). These correlations should also be discussed in the Discussion if you wish to include them. It may well be particularly interesting to test for associations between telomere dynamics and measures of oxidative stress.

6) This study includes a longitudinal data subset from Pelletier et al. 2023 (doi.org/10.3389/fevo.2023.1108293), which compared several of the same variables with partnership status across 2017-2019, and large parts of the text is consequently somewhat repetitive of that study. However, it should be discussed or clearly summarized if the results are consistent with the larger(?) dataset in Pelletier et al. 2023.

L41: Telomere length, not “telomeres”.
L42: Add “potential” to these survival and fitness effects because that has not been shown in this study.
L70-84: I suggest emphasizing in the Introduction when you are discussing comparisons and trade-offs across versus within species.
L77-84: These long-lived sea birds also show contrasting telomere dynamics compared to the “small short-lived passerines” outlining the expected trade-offs (references not included to avoid self-citations).
L80: Long-lived *sea birds.
L89: Add species name.
L126-130: Which three years were the subject of this study and how did divorce rate and breeding success vary across those years and across the subset of individuals actually included in this study? In other words, is the dataset representative of the colony?
L127: I do not think that the “704 pairs within 184 nests” is relevant to this study.
L143: Were there cases where birds acquired a new partner due to the death of a previous partner, and were such cases then excluded?
L228: “blood cells’ average telomere length”, not “average blood cells telomere length”?
L231: Did you include non-target controls?
L242: Delete hyphen in 2-primers.
L260: How many plates? Did you try to account for plate identity in the models including telomere length? See also Eisenberg 2016 doi.org/10.1093/ije/dyw191 for an alternative to reporting interassay repeatability as CV.
L281: Were partnership status and year correlated?
L284-286: This sentence is unclear to me – please elaborate which and what type of regressions you extracted residuals from. Instead of taking the residuals, could you simply include these variables as predictors in you models to directly control for them?
L285: What is included in “every biomarker” (e.g. including nutritional status and telomere length)?
L317: The Results section is a long read. I suggest an initial summary listing which variables varied with partnership status and year.
L325: *mass.
L347: Instead of writing “Figure 1 mean ± 95% CI did not include zero”, I suggest reporting the mean ± 95% CI.
L405: Were there sex differences in TL or TROC? Furthermore, did you try to account for (expected) effects of age or time on TL?
L408-411: “(TROC) was associated with breeding success, partnership status and year” followed by “TROC did not vary with partnership status” seems contradictory. Judging from Table S1, the first statement is wrong.
L422: I suggest structuring the Discussion similarly to how the Methods and Results outlines the variables measured and analyzed.
L438: “global changes” or simply environmental changes?
L455: Reformulate “gannets may shorten their telomeres” to something with less intentionality.
L457-458: See also e.g. Benowitz-Fredericks et al. 2022, doi.org/10.1098/rspb.2022.0139.
L464: For a more extensive meta-analysis of the association between TL and oxidative stress, see Armstrong & Boonekamp 2023, doi.org/10.1016/j.arr.2023.101854.
L467: For such a mechanism, see e.g. Haussmann et al. 2007, doi.org/10.1016/j.exger.2007.03.004.
L496: What does it mean that “sex has not been included in most models” – that sex was not in the best models?
L502: What do you mean by “global” environmental conditions?
L518: Stier et al. 2019 is missing from the reference list.
L581-586: This seems rather speculative. There appears to be no associations between TL and GLU in your study, which does not support your explanation?
L591: “food availability” was not included in this study, so I suggest reformulating this.
L592: Reformulate “long-term scale” to “three years”.
L596: Reformulate “during unfavorable years” to “during an unfavorable year”.
L619: Perhaps you should not thank reviewers for helpful comments in advance before you know whether those comments were helpful.

·

Basic reporting

In this study, Pelletier et al. investigated the physiological costs of reproduction and considered how the strategy of switching partners might alleviate or strengthen reproductive costs under varying degrees of environmental harshness in a long-lived seabird, the northern gannet. They considered an impressive number of physiological markers, including markers of inflammation, nutritional and oxidative stresses and molecular markers of aging. While this study brings new information on the physiological costs of reproduction and is valuable to the field, especially considering its very broad approach of the notion of physiological costs, I have some concerns regarding the sampling design and questions about the methods, which I believe should be addressed before the study can be published.

Experimental design

I believe that it should be specified from the beginning that this study is done focusing on three years (2017, 2018, 2019). As it is, we read that it is part of a long-term monitoring since 2008 and it is not clear before seeing the figures that only three years were used. This also brings some questions regarding the conclusions, as you actually have one good year, one medium year, and one bad year, which makes N=1 per category of year, and, although this provides a continuum, it could be argued the you are solely observing a “year effect” and that more years per category (good, medium, bad, Figure 3) would be needed to make any conclusion regarding the effect of the harshness of the environment, as there are many (biotic and abiotic) parameters that can vary between those years.

Methodology on qPCR to assess telomere length

L149-155: how long did it take before the samples were transferred to -80? Was this duration homogenized between samples/sampling sessions? This is especially important for telomere measurements as variation in the duration between sampling and freezing, as well as the conditions/matrix in which the samples are conserved can incur a lot of variation (Nussay et al. 2014).

L231-238: Beyond the quantity of DNA, did you also look at DNA quality and purity of the samples (using gel migration and the Nanodrop. See e.g. Viblanc et al 2020, Mol. Ecol., Tissier et al. Mol. Ecol. 2022)? This is essential when running qPCR analyses as degraded DNA and/or samples with reduced purity (purity is given by the absorbance at 230 nm, i.e. the ratio 230/260) can yield biased estimations of relative telomere length. Exclusion criteria are generally fixed depending on both the concentration and purity of the samples, i.e. concentration of 10ng/μL, and ratios at 1.7-2.0 (260/280) and >1.5 (230/260) are generally considered acceptable, but this can vary depending on the concentration/quality/purity.

L245: how did you chose to use 20 ng of DNA per reaction (this is relatively high compared to most studies on telomeres, which uses between 2.5 and 10 ng per reaction)?
L248-261: what were the Intraclass Correlation Coefficients (ICC) inter and intra-runs (obtained by repeating samples within a plate and on different plates)? These are essentials to assess the validity of the qPCR and should remain above 0.9. Also, how many plates did you have for this qPCR analysis?

Validity of the findings

Most analyses are convincing, and except for telomeres analyses for which I would need more information to assess the validity of the findings, I am convinced for other markers.

However, one comment that I could make regarding the finding is that you have a lot of physiological markers and, although analyzing them individually brings interesting information, it is also harder to get the full picture. Why not conducting an analysis to reduce the number of dimensions and/or try to connect them all together, even if only visually (i.e. a PCA, a PCoA or a path-analysis)? I believe this was perhaps the goal of Figure 3 but I am not sure to understand everything in Figure 3, including some logos used.

Regarding figure 2: the logos are hard to see on the figures and maybe not essential. In general, the quality of the figures is unfortunately not very high, which impairs the reading and assessment. This is maybe due to the pdf compression though and could be solved by submitting larger pictures when resubmitting.

Discussion

L444 – It is surprising to start this section discussing your results on telomeres with the (debated) notion of individual quality, which was not previously mentioned in your manuscript. Moreover, you do not discuss the potential seasonal variation in telomere length recorded in several species and conclude that the attrition recorded is detrimental for the birds' longevity, although it could be a “non-permanent state” associated with harsh or specific environmental conditions, life-history strategies or physiological sates. See for instance Criscuolo et al. 2020 Oecologia, Viblanc et al. 2022 Oecologia, Tissier et al. 2022 Molecular Ecology. Although these studies were conducted on mammals, telomere elongation or a positive relation between age and telomere length have also been recorded in some birds (e.g. including long-live seabirds in Haussman et al. 2003 Proc. R. Soc. B., see also Spurgin et al. 2017, J. Anim. Ecol) or in fish (Voituron et al. 2023 Proc. R. Soc. B, Panasiak et al. 2020 Genes). There is currently no consensus in the scientific literature regarding telomere elongation, but given the scope of your paper, this would be worth discussing.

Additional comments

Abstract
I believe the context section of the abstract would benefit from being detailed and more specific. Why is the lifestyle of the gannet challenging? Why do you mean by little safety margins? Why are feeding conditions challenging during reproduction?

Introduction
Overall, I believe that the notion of physiological costs of reproduction and the importance of studying them and understanding their role in shaping life-history trajectories, as well as the current debate on such costs in ecology, could be better introduced. I give some specific recommendations below.

Detailed comments
L35-35: in individuals who changed partners during unfavorable conditions or in all birds?

Intro
L54-55: why is it essential?
L58 – define oxidative stress at first mention.
L65-69 – repetitive wording; also, the cost of reproduction is debated beyond the type of markers used, and this should be introduced here, using some previous reviews and opinions or studies on the oxidative shielding hypothesis (e.g. Speakman and Garratt 2014, Metcalfe and Monaghan 2013, Viblanc et al. 2018)
L70-73: do you mean most of the work in birds?
L101-109: it should be specified here how many times each bird was sampled and at which period.
L108: in which tissue was telomere length assessed?
L135: recorded the body mass?
L214: maybe specify which information the assessment of the ALB/GLOB ration brings.
L283 – built
L444: a “.” is missing.

---

## Round 0.2 · Minor Revisions

We have received two additional evaluations of your work from the same previous reviewers. They both found your manuscript to be greatly improved overall. They suggested only a few additional minor revisions that will further improve the manuscript and that should be easy to integrate in the next version.

In summary they ask you to:

-include the code or raw data for all analyses/samples
-clarify if you included non-target controls
-clarify and justify your deltAICc threshold and adjust your conclusions accordingly
-clarify what happened with the samples between their freezing in the liquid nitrogen and their placement in a -80.
-rephrase/correct the text in some places

Reviewer 1 ·

Basic reporting

The authors have done a good job replying to all my comments and generally modified the manuscript accordingly. I have a few minor comments.

1) The authors have corrected a mistake in the calculation of telomere lengths, but this does not appear to change the main results. However, I suggest including the code or raw data whenever possible to mitigate such problems. Note that the calculated telomere lengths included are not raw data (but Ct thresholds would be).

Experimental design

2) The authors may have misunderstood my previous comment “L231: Did you include non-target controls?” – non-target controls do not refer to the golden sample, but are wells without DNA added, that are normally included on all plates to check for contamination. I asked this because I was wondering about the TL outliers (a problem the authors solved), but it is standard to report the inclusion of the controls.

Validity of the findings

3) It is now clearer that data dredging is used to find significant relationships between a range of physiological variables. This can be fine, but the authors appear to use a post hoc defined deltAICc threshold of 2.5 and I think this may because several of the candidate models including the variable of interest, partnership status, are outside the conventional threshold of 2 (e.g. model of TROC has deltaAICc = 2.49. However, in Fig. 1 you report a deltaAICc threshold of 2?). This may have resulted in formulations such as: “TROC was associated with breeding stage, partnership status and year (χ^2 = 4.23, df = 1, p 0.04), but varied only with year (t12 = -6.69, p < 0.0001) and breeding stage (t34 = 2.06, p = 0.047).” It does not make sense that TROC should be associated with but does not vary with partnership status. It should be clearer stated that model selection provided little support for the inclusion of partnership status in the model explaining variation in TROC, and it does not seem correct to conclude that TROC was “associated with” partnership status (L490 in manuscript with tracked changes). This also seems to occur with e.g. OHdG, HL, URIC, where you select models including partnership status although that variable does not really improve the models (or has any effect on the response variables). For simplicity, I would have preferred to choose the most parsimonious model in all cases – or alternatively skip model selection and construct models with your set of variables of interest.

Additional comments

L644: telomeres underwent what?
I like the revised discussion of telomere dynamics and the discussion of potential year effects – I have recently been involved in studies in sparrows where we find very large hatch year effects on telomere dynamics that manifest as cohort effects both in early-life (https://onlinelibrary.wiley.com/doi/10.1111/mec.16288) and later in life (https://www.nature.com/articles/s41598-023-31435-9, note self-citations) across twenty years. The variation due to year effects were larger than that due to all other environmental, parental or genetic factors included. These suggest, in line with the gannets, that individual heterogeneity in telomere dynamics is mainly driven by annual environmental stochasticity, which was somewhat surprising to me.

Title: Consider specifying “varying breeding seasons”, i.e. “environmentally varying”, or “varying breeding conditions” (it is not the season that varies).

·

Basic reporting

I have now reviewed the revised version of the manuscript Under pressure – Exploring partner changes, physiological responses and telomere dynamics in northern gannets across varying breeding seasons, by Pelletier et al. I believe the authors have done an excellent work addressing all the comments and questions initially raised. The addition of methodological information will ensure the reproducibility of the paper. I particularly appreciate the extensive work on the abstract, introduction and first section of the discussion to clarify the message. I am enthusiastic to see the manuscript in its current form and believe that it is suitable for publication. I only have very minor comments (please see below).

Experimental design

Methods
I appreciate all the details added in the methods, including on telomeres analyses. I have one last question though: on L246-247 you write “From the centrifuged tube, plasma and blood cells phases were divided in cryotubes and were stored immediately in liquid nitrogen (-196 °C), and subsequently (after 6-7 days) in a -80 °C freezer until biomarkers analyses”

But what did you do with the samples for 6-7 days (between their freezing in the liquid nitrogen and their placement in a -80?). You mention it in the response to reviewers but it should be stated in the manuscript as well – “We kept them in this liquid for up to 6-7 days and sorted them on dry ice (-78 °C) before storing them in a -80 °C freezer”.

Validity of the findings

I am convinced by the validity of the findings.

Additional comments

Additional minor comments

Consider rephrasing:
L24 –the risk of reproductive failure? Or reproductive failure’s risk?
L33 – Than those that retained partners. Or than individuals that retained partners.
L34 – individuals that changed mates (instead of those)
L152 – Remove “also” and add “In addition” at the beginning of the sentence
L394 – In this study (instead of “for our investigation”)
L609 – something is missing – marked by?
L629 – between breeding seasons or during/within/throughout the breeding season?
L646 – telomere dynamics (not attrition) – this includes telomere elongation as well.
L653 – between these two groups.
L660 – a favorable breeding year stabilizes telomere length (not plural since you do not have several good years in the dataset)

Typos
L22 – “their” is missing twice (their foraging range and diversify their diet)
L140, 645 – space missing
L304-305 – replace the commas with periods (numbers)
L605 – pairing behavior is written twice
L608 – an improved

---

## Round 0.3 · accepted · Accept

I am satisfied with the last revisions made to the manuscript.